# Comparison of survival times of advanced cancer patients with palliative care at home and in hospital

**Jun Hamano**[1]*, **Ayano Takeuchi**[2], **Masanori Mori**[3], **Yasuhiro Saitou**[4],
**Takahide Yamaguchi**[5], **Nobuyuki Miyata**[6], **Masakatsu Shimizu**[7], **Ryo Yamamoto**[8],
**Yousuke Kimura**[9], **Yoshiyuki Kamiyama**[10], **Yasuyuki Arai**[11], **Hiroshi Matsuo**[12],
**Hideki Shishido**[13], **Kazushi Nakano**[14], **Tomohiro Nishi**[15], **Hiroka Nagaoka**[1],
**Naosuke Yokomichi**[3], **Isseki Maeda**[3,16], **Takashi Yamaguchi**[17], **Tatsuya Morita**[18],
**Takuya Shinjo**[19]

1 Faculty of Medicine, Department of Palliative and Supportive Care, University of Tsukuba, Tsukuba, Japan,
2 Department of Preventive Medicine and Public Health, School of Medicine, Keio University, Tokyo, Japan,
3 Division of Palliative and Supportive Care, Seirei Mikatahara General Hospital, Hamamatsu, Japan, 4 GP
Clinic Jiyugaoka, Tokyo, Japan, 5 Ohisama Medical Corporation, Ohisama Clinic, Hyogo, Japan, 6 Miyata
Clinic, Chikusei, Ibaraki, Japan, 7 Shimizu Medical Clinic, Akashi, Hyogo, Japan, 8 Saku Central Hospital
Advanced Care Center, Saku-shi, Nagano, Japan, 9 Yamato Clinic, Sakuragawa, Ibaraki, Japan,
10 Okinawa Chubu Hospital, Uruma, Okinawa, Japan, 11 Iki-iki Clinic, Yuki, Ibaraki, Japan, 12 Marguerite
Clinic, Nagoya, Aichi, Japan, 13 Shishido Internal Medicine Clinic, Sakura, Chiba, Japan, 14 Nakano
Zaitakuiryou Clinic, Ishiki, Kagosima, Japan, 15 Kawasaki Municipal Ida Hospital, Kawasaki, Kanagawa,
Japan, 16 Department of Palliative Care, Senri-chuo Hospital, Osaka, Japan, 17 Department of Medicine,
Division of Palliative Care, Konan Medical Center, Kobe, Japan, 18 Division of Palliative and Supportive
Care, Palliative Care Team, and Seirei Hospice, Seirei Mikatahara General Hospital, Hamamatsu, Shizuoka,
Japan, 19 Shinjo-clinic, Kobe, Hyogo, Japan

* junhamano@md.tsukuba.ac.jp

pone.0284147

**Data Availability Statement:** Data cannot be
shared publicly because of ethical aspects. For
more information please contact the IRB of both
studies. EASED study: mk-rinri@sis.seirei.or.jp

## Abstract

### Objectives

One primary concern about receiving care at home is that survival might be shortened
because the quality and quantity of treatment provided at home will be inferior to that given
in the hospital. Although our previous study demonstrated a longer survival of those with
home-based palliative care (PC), it lacked adjustment for some potential confounders
including symptoms and treatments during the stay. We aimed to compare the survival
times among advanced cancer patients receiving home-based and hospital-based PC with
adjusting for symptoms and treatments.

### Method

We compared survival time of participants who enrolled two multicenter, prospective cohort
studies of advanced cancer patients at 45-home-based PC services between July 2017 and
December 2017, and at 23-hospital-based PC services between January 2017 and December 2017. We analyzed with stratification by the estimated survival of Days, Weeks, and
Months, which were defined by modified Prognosis in Palliative care Study predictor

Come Home study: sien.ningenss@un.tsukuba.ac.
jp.

**Funding:** Funding Source This EASED study was
supported in part by a Grant-in-Aid from the
Japanese Hospice Palliative Care Foundation. This
COME Home study was supported in part by JSPS
KAKENHI Grant Number 19K10551 and
22H03305. The sponsor played no role in the study
design, collection, analysis, or interpretation of the
data, writing of the report, or in the decision to
submit the paper for publication.

**Competing interests:** None of the authors have
any financial or personal relationships to declare.

models-A. We conducted a Cox regression analysis with adjusting for potential confounders
including symptoms and treatments during the stay.

## Results

A total of 2,998 patients were enrolled in both studies and 2,878 patients were analyzed;
988 patients receiving home-based PC and 1,890 receiving hospital-based PC. The survival
time of patients receiving home-based PC was significantly longer than that of patients
receiving hospital-based PC for the Days Prognosis (estimated median survival time: 10
days [95% CI 8.1–11.8] vs. 9 days [95% CI 8.3–10.4], p = 0.157), the Weeks prognosis (32
days [95% CI 28.9–35.4] vs. 22 days [95% CI 20.3–22.9], p < 0.001), and the Months Prog-
nosis, (65 days [95% CI 58.2–73.2] vs. 32 days [95% CI 28.9–35.4], p < 0.001).

## Conclusion

In this cohort of advanced cancer patients with a Weeks or Months prognosis, those receiv-
ing home-based PC survived longer than those receiving hospital-based PC after adjusting
for symptoms and treatments.

## Introduction

Receiving care and dying in one's preferred place is an essential factor for a high quality of
death and dying [1,2]. More than half of all people would prefer to be cared for and die at
home, and the quality of death and dying are actually superior at home than in hospital [3–6].
Previous studies indicated greater pain intensity and non-home death preference predicted
hospital death [7] and 68% of bereaved families in Japan preferred for the patient to die at
home [8]. A qualitative study in Norway revealed that the preference to die at home was stable
over time and did not change with deterioration in health status and progression in illness [9].
However, receiving care at home may be achieved only at a very late disease stage or death at
home may not be achieved for multiple reasons such as insufficient health care resources, lack
of caregivers, and unpreparedness of the patient and family [4,10]. One of major concern is
that the quality and quantity of medical treatment provided at home will be inferior to that
given in hospital and that survival might be shortened [11–13].

A small retrospective study revealed that cancer patients receiving home-based palliative
care had significantly longer survival times than those in hospital, though it was lacked of
adjustment for sufficient prognostic factors and was conducted at a single center with a small
number of patients [14].

Then we conducted the secondary analysis of a multicenter, prospective cohort study which
demonstrated that cancer patients who died receiving home-based palliative care had similar
or longer survival times than cancer patients who died in hospital with specialized palliative
care. However, our study did not adjust for symptom severity and medical treatments during
care that would influence survival [15], whether the survival time is actually different accord-
ing to the type of palliative care remains unclear. In addition, the evidence of the difference in
medical treatment between palliative care at home and in hospital, which might be used to
interpret the difference of survival times, is lacking. Thus, we thought it is needed to explore
the difference of the survival time between advanced cancer patients receiving home-based

palliative care and hospital-based palliative care with considering the symptom severity and medical treatments during care.

The present study mainly aimed to investigate the potential difference of the survival time between advanced cancer patients receiving home-based palliative care and hospital-based palliative care, adjusting not only for patient characteristics and prognostic factors, but also for symptoms and treatments.

## Materials and methods

We conducted two multicenter, prospective cohort studies of advanced cancer patients who were receiving home-based palliative care (Come Home study) or hospital-based palliative care (EASED study) in Japan, to compare the difference in survival time. Both studies aimed to document and account for the symptoms and medical treatments experienced by advanced cancer patients at the end of life. We standardized survey items and assessment tools. The Come Home study was conducted at 45 home care services between July 2017 and December 2017, and the EASED study was conducted at 23 hospital-based palliative care centers between January 2017 and December 2017. The home-based palliative care is a part of standard clinical care in Japan.

The physician primarily responsible for each patient performed an evaluation and recorded all outcome measures on the day of enrolment, and followed the patient until death or six months after enrollment. The physician evaluated the patients at least once a day in hospital, and at least once a week, and often daily, at home. The evaluation period ended when home care ended for reasons other than death or patients were discharged from hospital alive within six months. The patients in hospital-based PC usually experience an outpatient clinic, admission to a hospital, discharge to community palliative care, and die in a hospital. Both studies were conducted in accordance with the ethical standards of the Declaration of Helsinki and the ethical guidelines for research presented by the Ministry of Health, Labour, and Welfare of Japan. The institutional review boards of all participating services approved this study and main institutional review boards (Come Home study: University of Tsukuba, EASED study: Seirei Mikatahara General Hospital) approved the use of existing EASED data for secondary analysis, and to combine the data for analysis. The authors did not access to information that could identify individual participants during or after data collection.

### Patients

We enrolled eligible patients separately for both studies. The Come Home study enrolled patients when they started home-based palliative care, and the EASED study enrolled the patients when they started hospital-based palliative care at the participating facilities during the study period. The eligibility criteria for the two studies were the same; 1) 18 years old or older, 2) locally advanced or metastatic cancer (including hematopoietic neoplasms), and 3) started home-based or hospital-based palliative care at the participating facilities. Both studies excluded all patients who declined to participate in the study. Patients scheduled to be transferred or discharged within a week were excluded from receiving hospital-based palliative care.

### Outcomes

The primary endpoint of this study was survival time of participants. Survival time was defined as the period from the day of enrolment to the date of death. To adjust for background factors with a potential influence on survival time, we obtained data on the day of enrolment to formulate modified Prognosis in Palliative care Study predictor models-A (PiPs-A) [16], and data

to formulate the Age-adjusted Charlson Comorbidity Index (ACCI) [17] which includes age and comorbidities. The modified PiPS-A includes the following: site of primary cancer, metastatic site, Abbreviated Mental Test score by physician rating, heart rate, anorexia, dyspnea, dysphasia, fatigue, weight loss in the previous month, Eastern Cooperative Oncology Group performance status, and global health status (rated on a specific 7-point scale used in the original study: [1] extremely poor health to [7] normal health). Symptoms were recorded as being either present or absent. Cognitive status was evaluated according to the Abbreviated Mental Test score used in the original Prognosis in Palliative Care Study models, as reported by Gwilliam et al [18]. In the current study, cognitive status was rated as absent if the score on the Abbreviated Mental Test was 4 points or more, and as present if the score was less than 3 (scoring was performed by a physician without interviewing the patient).

We also obtained data on the day of enrolment to formulate the Palliative Prognostic Index (PPI) [19] as a covariate factor of survival time. The PPI includes the following; Palliative Performance Scale (categorized into three groups: 10–20, 30–50, and 60 or more), oral intake (categorized as severely reduced, moderately reduced, or absent), edema (categorized as present or absent), dyspnea at rest (present or absent), and delirium defined by Diagnostic and Statistical Manual of Mental Disorders 5 (DSM-5) (present or absent).

We recorded several other symptoms and medical treatment factors associated with survival time based on the previous study and discussion among the researchers [20–24].

The symptoms on the day of enrolment, including pleural effusion (categorized as present or absent), asities (present or absent), bowel obstruction (present or absent), and hyperactive delirium defined by Memorial Delirium Assessment Scale item 9 (MDAS#9) [25] (present or absent) were recorded along with the symptom severity defined by the Integrated Palliative Care Outcome Scale (IPOS) and scored as 0 (not at all), 1 (slight), 2 (moderate), 3 (severe), or 4 (overwhelming), and the prevalence for any IPOS symptoms scored as 2, 3 or 4 [26,27]. We also recorded the symptoms during the enrolled periods; delirium defined by DSM-5, hyperactive delirium defined by MDAS#9, and the symptom severity at one week, three days, and one day before death (i.e., weakness or lack of energy at three days before death).

Medical treatment at the day of enrolment, such as chemotherapy within a month, use of oxygen therapy, use of any catheter, opioid dosage, and use of antipsychotic drugs, was recorded. Similarly, we recorded the medical treatment during the enrolled periods; use of antipsychotic drugs, palliative sedation, and medical treatment before death (i.e., dosage of opioids at one week before death and parenteral hydration during the 24 hours before death).

## Statistical analysis

We excluded patients whose date of death was missing, but we did not exclude or separate patients who moved from one setting to another. We conducted our analyses on complete-cases and then applied multivariate multiple imputations by chained equations (MICE) with patients having missing data before death, namely patients who moved from home to other settings, to compare survival times.

We performed MICE 10 times for the variables on the day of enrolment, i.e., use of oxygen therapy, three categories of modified PiPs-A, PPI≥6.5, presence of delirium, presence of hyperactive delirium, IPOS of dyspnea, fatigue, bowel obstruction, and for the variables during the enrolled periods, i.e., presence of delirium, presence of hyperactive delirium, palliative sedation, and use of antibiotics during the 24 hours before death. We also confirmed beforehand that the distribution and proportions of the variables on the day of enrolment did not change between subjects who were supplemented by MICE and those who were not.

To explore the difference in survival times among patients with similar background factors having a potential influence on survival time, we selected a set of confounders between the settings of PC (receiving home-based PC or hospital-based PC) and outcome (survival from the day of enrollment) based on several previous studies and clinical knowledge [20–24]. Subsequently, we compared survival times for the following modified PiPS-A survival groups: patients surviving for Days (0–13 days), Weeks (14–55 days), and Months (>55 days), based on our previous study [15]. We plotted survival curves with the Kaplan-Meier method and compared the survival times of patients who were receiving home-based palliative care and hospital-based palliative care. In addition, we conducted a Cox regression analysis to estimate the adjusted hazard ratio (HR) for survival of patients receiving home-based versus hospital-based PC by adjusting the confounders. We defined the confounders based on previous studies and discussion among the authors [19,28–33]; age (per decade), sex, primary cancer site, presence of metastasis, chemotherapy within a month, PiPs-A, PPI ($\geq$6.5), ACCI ($\geq$6), hyperactive delirium, use of oxygen therapy, use of any catheter, and symptoms and treatment at enrollment and during care. We defined the cutoffs of opioid dosage as greater than or equal to 120 mg/day based on a previous study that explored whether opioids influenced survival among advanced cancer patients [33].

We conducted sensitivity analyses using propensity score matching for patients' background data of home-based and hospital-based PC to assess how substantial the effect of missing (unknown) confounders masked a true HR about the confounders of survival time. Differences in variable distributions between home-based PC and hospital-based PC were compared using Student's t test for continuous variables and Pearson's $\chi$2 test or Fisher's exact test for categorical variables. Significance was accepted at P < .05 and analyses were conducted using SPSS-J software (version 25.0; IBM, Tokyo, Japan) and SAS 9.4 (Cary, NC, USA).

## Results

In total, 2,998 patients were enrolled in both studies; 1,102 patients receiving home-based palliative care and 1,896 patients receiving hospital-based palliative care. Among them, 120 patients (home-based: 114, hospital-based:6) were excluded because the date of death was missing. Subsequently, 2,878 patients were analyzed; 988 patients receiving home-based palliative care and 1,890 patients receiving hospital-based palliative care (S1 Appendix).

The mean age was 72.5 years (95% CI: 72.1–73.0). The gastrointestinal tract/ hepatobiliary system and pancreas were the most frequent sites of primary cancer, followed by lung cancer. Almost one-third of patients had PPI $\geq$ 6.5, and approximately half were predicted to have a weekly prognosis by modified PiPS-A. The mean survival time was 35.2 days (95% CI: 33.6–36.7) for all patients; 51.0 days (95% CI: 47.5–54.4) for patients receiving home-based palliative care and 26.9 days (95% CI: 25.6–28.3) for patients receiving hospital-based palliative care (Table 1).

### Symptoms and treatment until death

Table 2 shows the prevalence of symptoms and treatment until death. The patients with hospital-based palliative care had significantly poorer performance status health and higher symptom prevalence than patients in home-based palliative care, except for drowsiness, sore or dry mouth, hyperactive delirium at three days before death, and ascites.

**Table 1.  Patient characteristics at enrollment.**

| | All patients (n = 2878) | | Home-based palliative care (n = 988) | | Hospital-based palliative care (n = 1890) | | |
|---|---|---|---|---|---|---|---|
| | N | % | N | % | N | % | p-value |
| Age ≥ 65 | 2251 | 78.2 | 794 | 80.4 | 1457 | 77.1 | 0.046 |
| Male sex | 1518 | 52.7 | 558 | 56.5 | 960 | 50.8 | 0.004 |
| Married | 1851 | 64.3 | 701 | 71.0 | 1150 | 60.8 | < 0.001 |
| Live with family | 2225 | 77.3 | 852 | 86.2 | 1373 | 72.6 | < 0.001 |
| Underage child | 118 | 4.1 | 44 | 4.5 | 74 | 3.9 | 0.553 |
| Site of primary cancer | | | | | | | 0.057 |
| Lung | 500 | 17.4 | 182 | 18.4 | 318 | 16.8 | |
| Gastrointestinal / Hepatobiliary and pancreas | 1380 | 47.9 | 494 | 50.0 | 886 | 46.9 | |
| Gynecological | 180 | 6.3 | 61 | 6.2 | 119 | 6.3 | |
| Urogenital | 220 | 7.6 | 79 | 8.0 | 141 | 7.5 | |
| Breast | 184 | 6.4 | 53 | 5.4 | 131 | 6.9 | |
| Others | 414 | 14.4 | 119 | 12.0 | 295 | 15.6 | |
| Metastatic site | | | | | | | |
| Anywhere | 2327 | 80.9 | 724 | 73.3 | 1603 | 84.8 | < 0.001 |
| Liver | 1066 | 37.0 | 337 | 34.1 | 729 | 38.6 | 0.023 |
| Bone | 713 | 24.8 | 213 | 21.6 | 500 | 26.5 | 0.005 |
| Lung | 981 | 34.1 | 274 | 27.7 | 707 | 37.4 | < 0.001 |
| Central nervous system | 372 | 12.9 | 109 | 11.0 | 263 | 13.9 | 0.030 |
| Chemotherapy within a month | 370 | 12.9 | 198 | 20.0 | 172 | 9.1 | < 0.001 |
| Oxygen therapy | 704 | 24.5 | 136 | 13.8 | 568 | 30.1 | < 0.001 |
| Use of any catheter | 547 | 19.0 | 91 | 9.2 | 456 | 24.1 | < 0.001 |
| Pain IPOS[*1] ≥ 2 | 1023 | 35.5 | 359 | 36.3 | 664 | 35.1 | 0.902 |
| Shortness of breath IPOS[*1] ≥ 2 | 590 | 20.5 | 210 | 21.3 | 380 | 20.1 | 0.884 |
| Weakness or lack of energy IPOS[*1] ≥ 2 | 1198 | 41.6 | 413 | 41.8 | 785 | 41.5 | 0.377 |
| Drowsiness IPOS[*1] ≥ 2 | 594 | 20.6 | 181 | 18.3 | 413 | 21.9 | 0.004 |
| Sore or dry mouth IPOS[*1] ≥ 2 | 500 | 17.4 | 139 | 14.1 | 361 | 19.1 | < 0.001 |
| Anorexia | 2368 | 82.3 | 818 | 82.8 | 1550 | 82.0 | 0.644 |
| Dysphagia | 823 | 28.6 | 201 | 20.3 | 622 | 32.9 | < 0.001 |
| Weight loss in the previous month | 2159 | 75.0 | 779 | 78.8 | 1380 | 73.0 | < 0.001 |
| Edema | 1243 | 43.2 | 374 | 37.9 | 869 | 46.0 | < 0.001 |
| Pleural effusion | 748 | 26.0 | 194 | 19.6 | 554 | 29.3 | < 0.001 |
| Ascites | 851 | 29.6 | 284 | 28.7 | 567 | 30.0 | 0.492 |
| Bowel obstruction | 355 | 12.3 | 99 | 10.0 | 256 | 13.5 | 0.007 |
| Delirium (DSM[*2]-V) | 677 | 23.5 | 95 | 9.6 | 582 | 30.8 | < 0.001 |
| Hyperactive delirium (MDAS[*3] item 9) | 367 | 12.8 | 61 | 6.2 | 306 | 16.2 | < 0.001 |
| Abbreviated Mental Test by physician rating ≤ 3 | 842 | 29.3 | 170 | 17.2 | 672 | 35.6 | < 0.001 |
| ECOG PS[*4] | | | | | | | < 0.001 |
| 0–1 | 129 | 4.5 | 105 | 10.6 | 24 | 1.3 | |
| 2 | 377 | 13.1 | 220 | 22.3 | 157 | 8.3 | |
| 3 | 1167 | 40.5 | 372 | 37.7 | 795 | 42.1 | |
| 4 | 1205 | 41.9 | 291 | 29.5 | 914 | 48.4 | |
| Global Health | | | | | | | < 0.001 |
| 1: markedly poor | 325 | 11.3 | 72 | 7.3 | 253 | 13.4 | |
| 2 | 746 | 25.9 | 225 | 22.8 | 521 | 27.6 | |
| 3 | 1133 | 39.4 | 373 | 37.8 | 760 | 40.2 | |

(*Continued*)

**Table 1.** (Continued)

| | All patients (n = 2878) | | Home-based palliative care (n = 988) | | Hospital-based palliative care (n = 1890) | | |
|---|---|---|---|---|---|---|---|
| | N | % | N | % | N | % | p-value |
| 4 | 457 | 15.9 | 184 | 18.6 | 273 | 14.4 | |
| 5–7: normal health | 215 | 7.5 | 134 | 13.6 | 81 | 4.3 | |
| PiPs-A[*5] | | | | | | | < 0.001 |
| modified PiPS-A: Months | 509 | 17.7 | 258 | 26.1 | 251 | 13.3 | |
| modified PiPS-A: Weeks | 1428 | 49.6 | 535 | 54.1 | 893 | 47.2 | |
| modified PiPS-A: Days | 908 | 31.5 | 186 | 18.8 | 722 | 38.2 | |
| Palliative Prognostic Index ≥ 6.5 | 905 | 31.4 | 167 | 16.9 | 738 | 39.0 | < 0.001 |
| Age-adjusted Charlson comorbidity index ≥ 6 | 2753 | 95.7 | 926 | 93.7 | 1827 | 96.7 | < 0.001 |
| Opioid dosage (OME[*6] ≥ 120 mg/day) | 284 | 9.9 | 64 | 6.5 | 220 | 11.6 | < 0.001 |
| Using antipsychotic drug | 526 | 18.3 | 99 | 10.0 | 427 | 22.6 | < 0.001 |
| | Mean | 95% CI | Mean | 95% CI | Mean | 95% CI | |
| Age (yrs) | 72.5 | 72.1–73.0 | 72.8 | 72.1–73.6 | 72.4 | 71.8–72.9 | 0.333 |
| Age-adjusted Charlson comorbidity index | 10.8 | 10.7–10.9 | 10.5 | 10.3–10.6 | 11.0 | 10.8–11.1 | < 0.001 |
| Palliative Prognostic Index | 5.4 | 5.3–5.5 | 4.1 | 3.9–4.3 | 6.1 | 5.9–6.2 | < 0.001 |
| Opioid dosage per day (OME, mg/day) | 41.9 | 36.1–47.8 | 39.1 | 23.6–54.6 | 43.4 | 39.6–47.3 | 0.592 |

[*1]IPOS: Integrated Palliative Care Outcome Scale.

[*2]DSM: Diagnostic and Statistical Manual of Mental Disorders.

[*3]MDAS: Memorial Delirium Assessment Scale.

[*4]ECOG PS: Eastern Cooperative Oncology Group Performance Status.

[*5]PiPS-A: Prognosis in Palliative Care Study predictor model-A.

[*6]OME: Oral morphine equivalent.

## Comparison of survival time between home-based palliative care and hospital-based palliative care

The survival of patients receiving home-based palliative care was significantly longer than that of those receiving hospital-based palliative care for the Weeks prognosis group (estimated median survival time: 32 days [95% CI 28.9–35.4] vs. 22 days [95% CI 20.3–22.9], p<0.001) and the Months prognosis group (65 days [95% CI 58.2–73.2] vs. 32 days [95% CI 28.9–35.4], p<0.001), as defined by PiPs-A. No significant difference was identified in the Days prognosis group (10 days [95% CI 8.1–11.8] vs. 9 days [95% CI 8.3–10.4], p = 0.157) (Fig 1).

Cox proportional hazards analysis revealed that home-based palliative care had a significant positive influence on survival time in both the unadjusted (HR, 0.70 [95% CI, 0.64–0.77]; P< 0.001) and adjusted models with patients' background, symptoms, and treatment (at enrollment, during care, at 1 week before death), symptoms at 3 days before death, and treatment at last day before death (HR, 0.82 [95% CI, 0.71–0.95]; P = 0.007). High-dose opioids (oral morphine equivalent ≥ 120 mg) at one week before death had a significant positive influence on survival time, although that treatment at enrollment had a significant negative influence and that on the last day before death had no significant influence on survival time. Parental hydration at one week before death had a significant positive influence, whereas that on the last day before death had a significant negative influence (Table 3).

**Table 2. Symptoms and treatments until death.**

| | All patients (n = 2878) | | Home-based palliative care (n = 988) | | Hospital-based palliative care (n = 1890) | | |
|---|---|---|---|---|---|---|---|
| | N | % | N | % | N | % | p-value |
| ECOG PS*[1] | | | | | | | |
| 1 week before death | | | | | | | < 0.001 |
| 0–1 | 10 | 0.3 | 6 | 0.6 | 4 | 0.2 | |
| 2 | 76 | 2.6 | 44 | 4.5 | 32 | 1.7 | |
| 3 | 598 | 20.8 | 217 | 22.0 | 381 | 20.2 | |
| 4 | 1571 | 54.6 | 415 | 42.0 | 1156 | 61.2 | |
| 3 days before death | | | | | | | < 0.001 |
| 0–1 | 8 | 0.3 | 6 | 0.6 | 2 | 0.1 | |
| 2 | 20 | 0.7 | 13 | 1.3 | 7 | 0.4 | |
| 3 | 264 | 9.2 | 97 | 9.8 | 167 | 8.8 | |
| 4 | 2002 | 69.6 | 564 | 57.1 | 1438 | 76.1 | |
| last day before death | | | | | | | 0.042 |
| 0–1 | 5 | 0.2 | 3 | 0.3 | 2 | 0.1 | |
| 2 | 7 | 0.2 | 5 | 0.5 | 2 | 0.1 | |
| 3 | 96 | 3.3 | 28 | 2.8 | 68 | 3.6 | |
| 4 | 2200 | 76.4 | 646 | 65.4 | 1554 | 82.2 | |
| Weakness or lack of energy IPOS*[2] ≥ 2 | | | | | | | |
| 3 days before death | 984 | 34.2 | 336 | 34.0 | 648 | 34.3 | 0.012 |
| Drowsiness IPOS*[2] ≥ 2 | | | | | | | |
| 3 days before death | 703 | 24.4 | 267 | 27.0 | 436 | 23.1 | 0.356 |
| Sore or dry mouth IPOS*[2] ≥ 2 | | | | | | | |
| 3 days before death | 546 | 19.0 | 186 | 18.8 | 360 | 19.0 | 0.118 |
| Delirium (DSM*[3]-V) | | | | | | | |
| during the care | 910 | 31.6 | 211 | 21.4 | 699 | 37.0 | < 0.001 |
| Hyperactive delirium (MDAS*[4] item 9) | | | | | | | |
| during the care | 923 | 32.1 | 217 | 22.0 | 706 | 37.4 | < 0.001 |
| 3 days before death | 540 | 18.8 | 167 | 16.9 | 373 | 19.7 | 0.312 |
| Antipsychotic drug for delirium | | | | | | | |
| during the care | 887 | 30.8 | 112 | 11.3 | 775 | 41.0 | < 0.001 |
| Ascites | | | | | | | |
| 3 days before death | 338 | 11.7 | 97 | 9.8 | 241 | 12.8 | 0.796 |
| Fever 1 week before death | 752 | 26.1 | 131 | 13.3 | 621 | 32.9 | < 0.001 |
| Antibiotic use at last day before death | 156 | 5.4 | 11 | 1.1 | 145 | 7.7 | < 0.001 |
| Parenteral hydration | | | | | | | |
| 1 week before death | 1046 | 36.3 | 139 | 14.1 | 907 | 48.0 | < 0.001 |
| last day before death | 1050 | 36.5 | 131 | 13.3 | 919 | 48.6 | < 0.001 |

*(Continued)*

**Table 2.** (Continued)

| | All patients (n = 2878) | | Home-based palliative care (n = 988) | | Hospital-based palliative care (n = 1890) | | |
|---|---|---|---|---|---|---|---|
| | N | % | N | % | N | % | p-value |
| Opioid dosage (OME[*5] ≥ 120 mg/day) | | | | | | | |
| 1 week before death | 297 | 10.3 | 77 | 7.8 | 220 | 11.6 | < 0.001 |
| last day before death | 348 | 12.1 | 87 | 8.8 | 261 | 13.8 | < 0.001 |
| Palliative sedation | 206 | 7.2 | 45 | 4.6 | 161 | 8.5 | 0.007 |
| | Mean | 95% CI | Mean | 95% CI | Mean | 95% CI | |
| Opioid dosage per day (OME, mg/day) | | | | | | | |
| 1 week before death | 62.8 | 54.4–71.2 | 43.5 | 26.5–60.4 | 77.7 | 70.7–84.7 | < 0.001 |
| last day before death | 71.1 | 62.6–79.7 | 59.5 | 37.6–81.3 | 77.1 | 70.8–83.5 | 0.128 |
| Periods of palliative sedation (days) | 1.0 | 0.8–1.2 | 0.6 | 0.4–0.8 | 3.1 | 2.6–3.6 | < 0.001 |
| Survival (days; median) | 35.2 (21.0) | 33.6–36.7 | 51.0 (32.0) | 47.5–54.4 | 26.9 (17.0) | 25.6–28.3 | < 0.001 |
| Good death scale (0–15) | 12.0 | 11.9–12.2 | 12.2 | 12.0–12.4 | 12.0 | 11.8–12.1 | 0.048 |

[*1]ECOG PS: Eastern Cooperative Oncology Group Performance Status

[*2]IPOS: Integrated Palliative Care Outcome Scale.

[*3]DSM: Diagnostic and Statistical Manual of Mental Disorders

[*4]MDAS: Memorial Delirium Assessment Scale

[*5]OME: Oral morphine equivalent.

## Discussion

To the best of our knowledge, this is the first large-scale prospective, multicenter study to compare the survival time of advanced cancer patients receiving home-based palliative care or hospital-based palliative care adjusting for symptoms and treatment factors.

The most important finding of this study is that advanced cancer patients receiving home-based palliative care with an estimated Weeks or Months prognosis survived longer than those receiving hospital-based PC, whereas those with an estimated Days prognosis survived similar times receiving hospital-based palliative care after adjusting for symptoms and treatments. Our previous multicenter study suggested that cancer patients who died receiving home-based palliative care had similar or significantly longer survival times than those who died in hospital with specialized palliative care services, although that study lacked adjustment for symptoms and medical treatment. As several previous studies demonstrated that symptoms and treatment are associated with the survival time of advanced cancer patients [34–37], our current study was novel to present that advanced cancer patients receiving home-based palliative care with an estimated Weeks or Months prognosis survived longer than those receiving hospital-based palliative care by adjusting for symptoms and treatment factors.

One possible hypothesis that estimated Weeks or Months prognosis survived longer than those receiving hospital-based palliative care might be due to the difference in the living environment. In other words, the inpatient environment may interfere with the patient's autonomy and motivation, which in turn may negatively affect activities, appetite, and other matters

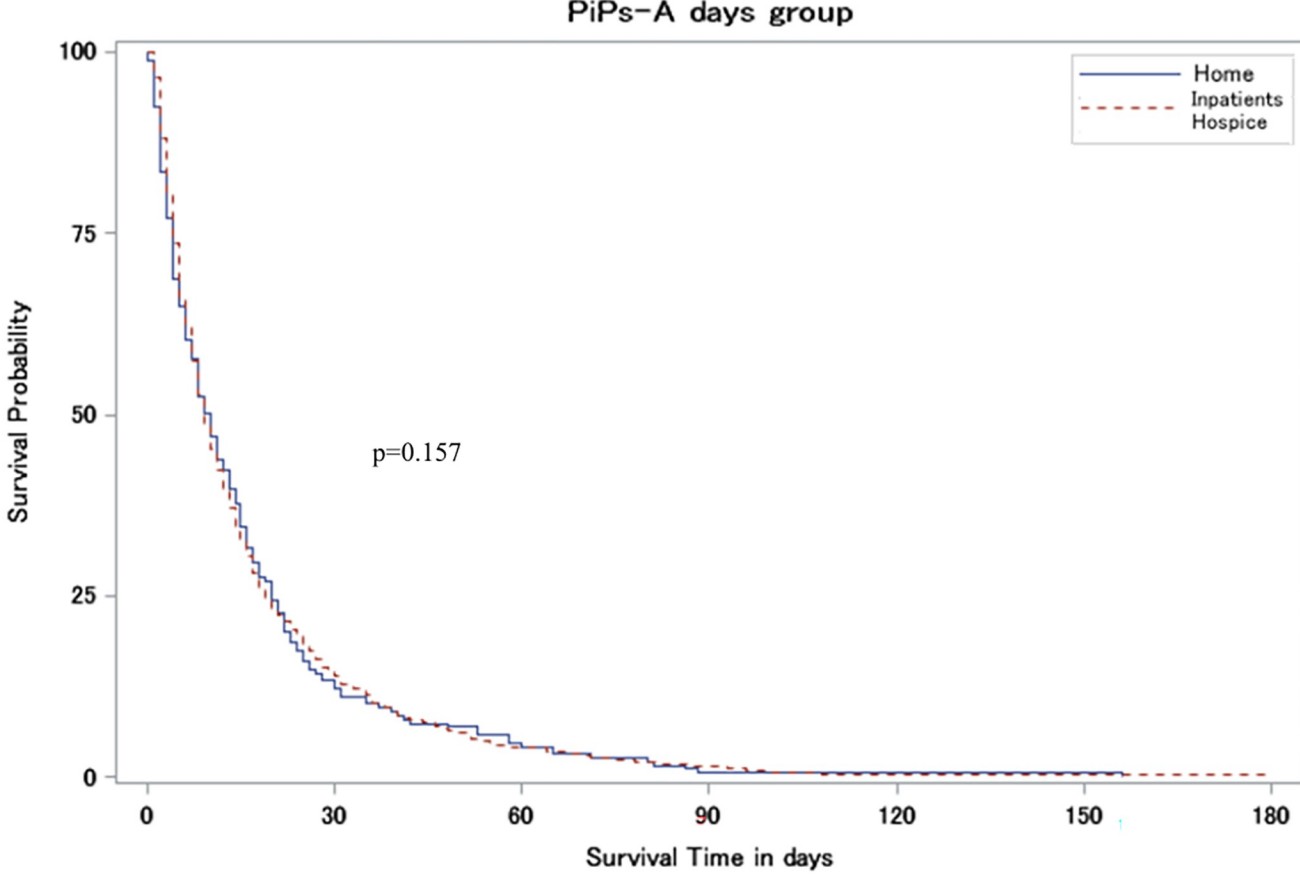

Estimated median survival time: 10 days [95% CI 8.1 – 11.8] vs. 9 days [95% CI 8.3-10.4], p=0.157

**Fig 1. Kaplan-Meier survival curves stratified by the place of care for 3 groups defined according to Prognosis in Palliative Care Study predictor model A (PiPs-A): Days' group (0–13 days), weeks' group (14–55 days), and months' group (≧56 days).**

necessary for survival. Another possible hypothesis that an estimated Days prognosis had similar survival times was the physiological effects of the dying process prevail over the type of specialist palliative care. This explanation would reflect the concept of palliative care; palliative care intends neither to hasten or postpone death and focus on relief from pain and other distressing symptoms. This result implied that distinguishing the Days prognosis is important to decide treatment and care plan.

Of note, our study did not indicate a clear association between the quality of symptom control or burden of suffering and survival time. However, previous studies demonstrated that symptoms are associated with the survival time of advanced cancer patients [34–37]. Therefore, one possible explanation was that our study could not reveal symptom control details with day-by-day assessment.

A strength of this study was the large sample size that used many adjustment variables based on multiple areas including patient characteristics, prognostic factors, symptoms, and medical treatments. Since it is ethically difficult to conduct RCT, our large-scale prospective, multicenter study with multifactorial adjustments would be the high level of evidence available [38]. Another strength of our study is that missing data caused by changes in the place of care were supplemented and analyzed with multivariate multiple imputations by chained equations.

**Table 3. Cox proportional hazards analysis of survival time.**

|  | Hazard Ratio | 95% CI | p-value |
|---|---|---|---|
| Unadjusted model |  |  |  |
| Home-based | 0.70 | 0.64–0.77 | < 0.001 |
| Adjusted model |  |  |  |
| Home-based | 0.82 | 0.71–0.95 | 0.007 |
| Age (per decade) | 0.96 | 0.91–1.01 | 0.125 |
| Female | 0.89 | 0.78–1.01 | 0.072 |
| Site of primary cancer: Others |  |  |  |
| Lung | 1.06 | 0.86–1.32 | 0.579 |
| Gastrointestinal | 1.07 | 0.89–1.30 | 0.466 |
| Gynecological | 1.06 | 0.81–1.50 | 0.702 |
| Urogenital | 1.03 | 0.73–1.27 | 0.813 |
| Breast | 1.46 | 1.00–1.84 | 0.015 |
| Chemotherapy within a month | 0.90 | 0.77–1.06 | 0.225 |
| Modified PiPS-A[*1]: Months |  |  |  |
| Modified PiPS-A: Weeks | 1.48 | 1.25–1.75 | < 0.001 |
| Modified PiPS-A: Days | 3.18 | 2.55–3.96 | < 0.001 |
| Palliative Prognostic Index $\geq$ 6.5 | 1.37 | 1.16–1.62 | < 0.001 |
| Age-adjusted Charlson comorbidity index $\geq$ 6 | 0.96 | 0.72–1.29 | 0.800 |
| Symptom and treatment at enrollment |  |  |  |
| Pleural effusion | 1.01 | 0.88–1.16 | 0.877 |
| Ascites | 1.66 | 1.42–1.94 | < 0.001 |
| Bowel obstruction | 1.06 | 0.88–1.28 | 0.542 |
| Pain IPOS[*2] $\geq$ 2 | 1.08 | 0.90–1.15 | 0.242 |
| Shortness of breath IPOS $\geq$ 2 | 1.04 | 0.92–1.27 | 0.606 |
| Weakness or lack of energy IPOS $\geq$ 2 | 1.44 | 1.20–1.57 | < 0.001 |
| Drowsiness IPOS $\geq$ 2 | 1.09 | 0.90–1.25 | 0.319 |
| Sore or dry mouth IPOS $\geq$ 2 | 1.33 | 1.09–1.51 | < 0.001 |
| Hyperactive delirium | 1.55 | 1.12–2.15 | 0.009 |
| Oxygen therapy | 1.31 | 1.13–1.54 | < 0.001 |
| Use of any catheter | 1.10 | 0.94–1.29 | 0.214 |
| Opioid dosage (OME[*3] $\geq$ 120 mg/day) | 1.35 | 1.08–1.68 | 0.007 |
| Using antipsychotic drug | 1.07 | 0.91–1.25 | 0.416 |
| Symptom and treatment during the care |  |  |  |
| Delirium | 0.97 | 0.84–1.10 | 0.620 |
| Using antipsychotic drug | 0.91 | 0.79–1.06 | 0.228 |
| Palliative sedation | 1.02 | 0.83–1.27 | 0.835 |
| Symptom and treatment at 1 week before death |  |  |  |
| Fever | 0.96 | 0.84–1.09 | 0.520 |
| Opioid dosage (OME $\geq$ 120 mg/day) | 0.60 | 0.45–0.80 | < 0.001 |
| Parenteral hydration | 0.81 | 0.66–0.99 | 0.040 |
| Symptom at 3 days before death |  |  |  |
| Weakness or lack of energy IPOS $\geq$ 2 | 1.07 | 0.93–1.22 | 0.342 |
| Drowsiness IPOS $\geq$ 2 | 0.77 | 0.67–0.88 | < 0.001 |
| Sore or dry mouth IPOS $\geq$ 2 | 0.95 | 0.82–1.09 | 0.451 |
| Ascites | 0.89 | 0.74–1.06 | 0.177 |
| Treatment at last day before death |  |  |  |
| Opioid dosage (OME $\geq$ 120 mg/day) | 1.11 | 0.86–1.43 | 0.437 |

(*Continued*)

**Table 3.** (Continued)

|  | Hazard Ratio | 95% CI | p-value |
|---|---|---|---|
| Parenteral hydration | 1.29 | 1.05–1.57 | 0.014 |
| Antibiotic use | 1.03 | 0.80–1.32 | 0.819 |

[*1]PiPS-A: Prognosis in Palliative Care Study predictor model-A.

[*2]IPOS: Integrated Palliative Care Outcome Scale.

[*3]OME: Oral morphine equivalent.

The current study also had some limitations. First, we were unable to control the general condition of the two groups at enrollment. We thought survival time cannot be compared without aligning the general condition among two groups, and it is difficult to achieve a rigorous alignment of the general conditions, even with maximum adjustment of the various prognostic markers. Therefore, our result, advanced cancer patients with a Weeks or Months prognosis, those receiving home-based palliative care survived longer than those receiving hospital-based palliative care, could be due to their better general conditions at enrollment. However, since we adjusted for the validated prognostic scales, PiPS-A and PPI, we do not believe that there was an extremely large difference in the general condition of the two groups.

Second, we were unable to adjust for residual confounding factors affecting the choice of the type of palliative care and survival time such as the preferences of patients and their families, family support, the details of dose-response treatment, and spiritual well-being.

Third, we were unable to consider the interaction of successive symptoms and medical treatments in terms of time-dependent manner. Thus, further research is needed to clarify the potential effects of symptoms and medical treatments as time-dependent confounding factors on survival time.

Fourth, sometimes a patient at home could not be assessed on a defined date such as a week before death, three days before death, or the last day before death. It is therefore possible that some of the evaluations at home were based on physicians' estimates, which may have caused under- or overestimation of symptoms for home-based palliative care patients. Nevertheless, we believe that our large-scale, prospective, multicenter study offers the highest available level of evidence and closely maps reality.

Fifth, we did not measure the number of patients with crossover between home and hospital parts of the study. While a high percentage of crossover may affect outcomes [39], we believe it did not seriously affect this study because very few of the facilities that participated in the two studies were located in the same area, and the study periods only partially overlapped.

Thus, we could not conclude a causal relationship or clarify the scientific mechanisms between the type of palliative of care and the survival time of advanced cancer patients in this observational study. Further observational studies based on causal inference are needed to uncover whether advanced cancer patients at home shorten their survival time compared to staying in the hospital [38]. Once this becomes clear, patients, families, and providers would feel more comfortable choosing home care if they knew that advanced cancer patients receiving home palliative care would not have a shorter survival time than those receiving inpatient palliative care.

## Conclusions

In this cohort of advanced cancer patients receiving home-based palliative care with an estimated Weeks or Months prognosis survived longer than those receiving hospital-based PC,

whereas those with an estimated Days prognosis survived similar times receiving hospital-based palliative care after adjusting for symptoms and treatments. Further studies are needed to clarify that the survival time of home patients is not shorter than that of inpatients.

## Supporting information

**S1 Appendix. Participant flow.**
(DOCX)

## Acknowledgments

This study was performed as part of the comparison of end-of-life trajectory in advanced cancer patients between home (Come Home study) and inpatient hospice (the East-Asian collaborative cross-cultural study to elucidate the dying process: EASED study).

The participating investigators and study sites of the Come Home study were as follows: Keijiro Miyake, M.D., Ph.D. (Keijiro Clinic), Manabu Tamura, M.D., Ph.D. (Osaka Home Healthcare Clinic), Junichiro Toya, M.D. (Sakura-shinmachi Urban Clinic), Hiroto Shirayama, M.D. (Osaka Kita Home Care Clinic), Takamichi Matsuki, M.D., Ph.D. (Fujisawa-Honmachi Family Clinic), Akihiro Ishikawa, M.D., (Ishikawa Rehabili Noushinkeigeka Clinic), Yasunori Muraoka, M.D. (Muraoka Home Clinic), Yasuhiro Saitou, M.D. (GP Clinic Jiyugaoka), Takahide Yamaguchi, M.D. (Yamaguchi Clinic), Tomohiro Nishi, M.D. (Kawasaki Municipal Ida Hospital), Nobuyuki Miyata, M.D. (Miyata Clinic), Masakatsu Shimizu, M.D., Ph.D. (Shimizu Medical Clinic), Ryo Yamamoto, M.D. (Saku Central Hospital Advanced Care Center), Yousuke Kimura, M.D. (Yamato Clinic), Yoshiyuki Kamiyama, M.D. (Okinawa Chubu Hospital), Yasuyuki Arai, M.D., Ph.D. (Iki-iki Clinic), Hiroshi Matsuo, M.D. (Margaret Clinic), Hideki Shishido, M.D. (Shishido Internal Medicine Clinic), Kazushi Nakano, M.D., Ph.D. (Nakano Zaitakuiryou Clinic), Kan Asahina, M.D. (Mutsumimachi Clinic), Maiko Haruki, M.D. (Orange Home-Care Clinic), Keiko Inoue, M.D. (Aisei Clinic), Sen Otomo, M.D., (Seimeikan Clinic).

The participating investigators and study sites in the EASED study were as follows: Satoshi Inoue, M.D. (Seirei Hospice, Seirei Mikatahara General Hospital), Naosuke Yokomichi, M.D., Ph.D. (Department of Palliative and Supportive Care, Seirei Mikatahara General Hospital), Hiroaki Tsukuura, M.D., Ph.D. (Department of Palliative Care, TUMS Urayasu Hospital), Toshihiro Yamauchi, M.D. (Seirei Hospice, Seirei Mikatahara General Hospital), Akemi Shirado Naito, M.D. (Department of palliative care Miyazaki Medical Association Hospital), Yu Uneno, M.D. (Department of Therapeutic Oncology, Graduate School of Medicine, Kyoto University), Akira Yoshioka, M.D., Ph.D. (Department of Oncology and Palliative Medicine, Mitsubishi Kyoto Hospital), Shuji Hiramoto, M.D. (Department of Oncology and Palliative Medicine, Mitsubishi Kyoto Hospital), Ayako Kikuchi, M.D. (Department of Oncology and Palliative Medicine, Mitsubishi Kyoto Hospital), Tetsuo Hori, M.D. (Department of Respiratory surgery, Mitsubishi Kyoto Hospital), Yosuke Matsuda, M.D. (Palliative Care Department, St.Luke's International Hospital), Hiroyuki Kohara, M.D., Ph.D. (Hiroshima Prefectural Hospital), Hiromi Funaki, M.D. (Hiroshima Prefectural Hospital), Keiko Tanaka, M.D., Ph.D. (Department of Palliative Care Tokyo Metropolitan Cancer & Infectious Diseases Center Komagome Hospital), Kozue Suzuki, M.D. (Department of Palliative Care Tokyo Metropolitan Cancer & Infectious Diseases Center Komagome Hospital), Tina Kamei, M.D. (Department of Palliative Care, NTT Medical Center Tokyo), Koji Amano, M.D. (Department of Palliative Medicine, Osaka City General Hospital), Teruaki Uno, M.D. (Department of Palliative Medicine, Osaka City General Hospital), Jiro Miyamoto, M.D. (Department of Palliative

Medicine, Osaka City General Hospital), Hirofumi Katayama, M.D. (Department of Palliative Medicine, Osaka City General Hospital), Hideyuki Kashiwagi, M.D., MBA. (Aso Iizuka Hospital / Transitional and Palliative Care), Eri Matsumoto, M.D. (Aso Iizuka Hospital / Transitional and Palliative Care), Takeya Yamaguchi, M.D. (Japan Community Health care Organization Kyushu Hospital / Palliative Care), Tomonao Okamura, M.D., MBA. (Aso Iizuka Hospital / Transitional and Palliative Care), Hoshu Hashimoto, M.D., MBA. (Inoue Hospital / Internal Medicine), Shunsuke Kosugi, M.D. (Department of General Internal Medicine, Aso Iizuka Hospital), Nao Ikuta, M.D. (Department of Emergency Medicine, Osaka Red Cross Hospital), Yaichiro Matsumoto, M.D. (Department of Transitional and Palliative Care, Aso Iizuka Hospital), Takashi Ohmori, M.D. (Department of Transitional and Palliative Care, Aso Iizuka Hospital), Takehiro Nakai, M.D. (Immuno-Rheumatology Center, St Luke's International Hospital), Takashi Ikee, M.D. (Department of Cardiology, Aso Iizuka Hospital), Yuto Unoki, M.D. (Department of General Internal Medicine, Aso Iizuka Hospital), Kazuki Kitade, M.D. (Department of Orthopedic Surgery, Saga-Ken Medical Centre Koseikan), Shu Koito, M.D. (Department of General Internal Medicine, Aso Iizuka Hospital), Nanao Ishibashi, M.D. (Environmental Health and Safety Division, Environmental Health Department, Ministry of the Environment), Masaya Ehara, M.D. (TOSHIBA), Kosuke Kuwahara, M.D. (Department of General Internal Medicine, Aso Iizuka Hospital), Shohei Ueno, M.D. (Department of Hematology / Oncology, Japan Community Healthcare Organization Kyushu Hospital), Shunsuke Nakashima, M.D. (Oshima Clinic), Yuta Ishiyama, M.D. (Department of Transitional and Palliative Care, Aso Iizuka Hospital), Akihiro Sakashita, M.D., Ph.D. (Department of Palliative Medicine, Kobe University School of Medicine), Hana Takatsu, M.D. (Division of Palliative Care, Konan Medical Center), Takashi Yamaguchi, M.D., Ph.D. (Division of Palliative Care, Konan Medical Center), Satoko Ito, M.D. (Hospice, The Japan Baptist Hospital), Toru Terabayashi, M.D. (Hospice, The Japan Baptist Hospital), Jun Nakagawa, M.D. (Hospice, The Japan Baptist Hospital), Tetsuya Yamagiwa, M.D., Ph.D. (Hospice, The Japan Baptist Hospital), Akira Inoue, M.D., Ph.D. (Department of Palliative Medicine Tohoku University School of Medicine), Takuhiro Yamaguchi, Ph.D. (Professor of Biostatistics, Tohoku University Graduate School of Medicine), Mitsunori Miyashita, R.N., Ph.D. (Department of Palliative Nursing, Health Sciences, Tohoku University Graduate School of Medicine), Saran Yoshida, Ph.D. (Graduate School of Education, Tohoku University), Yusuke Hiratsuka, M.D., Ph.D. (Department of Palliative Medicine Tohoku University School of Medicine), Keita Tagami, M.D., Ph.D. (Department of Palliative Medicine Tohoku University School of Medicine), Hiroaki Watanabe, M.D. (Department of Palliative Care, Komaki City Hospital), Takuya Odagiri, M.D. (Department of Palliative Care, Komaki City Hospital), Tetsuya Ito, M.D.,Ph.D. (Department of Palliative Care, Japanese Red Cross Medical Center), Masayuki Ikenaga, M.D. (Hospice, Yodogawa Christian Hospital), Keiji Shimizu, M.D., Ph.D. (Department of Palliative Care Internal Medicine, Osaka General Hospital of West Japan Railway Company), Akira Hayakawa, M.D., Ph.D. (Hospice, Yodogawa Christian Hospital), Rena Kamura, M.D. (Hospice, Yodogawa Christian Hospital), Takeru Okoshi, M.D., Ph.D. (Okoshi Nagominomori Clinic), Isseki Maeda M.D., Ph.D. (Department of Palliative Care, Senri-Chuo Hospital), Tomohiro Nishi, M.D. (Kawasaki Municipal Ida Hospital, Kawasaki Comprehensive Care Center), Kazuhiro Kosugi, M.D. (Department of Palliative Medicine, National Cancer Center Hospital East), Yasuhiro Shibata, M.D. (Kawasaki Municipal Ida Hospital, Kawasaki Comprehensive Care Center), Takayuki Hisanaga, M.D. (Department of Palliative Medicine, Tsukuba Medical Center Hospital), Takahiro Higashibata, M.D., Ph.D. (Department of General Medicine and Primary Care, Palliative Care Team, University of Tsukuba Hospital), Ritsuko Yabuki, M.D. (Department of Palliative Medicine, Tsukuba Medical Center Hospital), Shingo Hagiwara, M.D., Ph.D. (Department of Palliative Medicine, Yuai Memorial Hospital), Miho Shimokawa, M.

D. (Department of Palliative Medicine, Tsukuba Medical Center Hospital), Satoshi Miyake, M. D., Ph.D. (Professor, Department of Clinical Oncology Graduate School of Medical and Dental Sciences Tokyo Medical and Dental University (TMDU)), Junko Nozato, M.D. (Specially Appointed Assistant Professor, Department of Internal Medicine, Palliative Care, Medical Hospital, Tokyo Medical and Dental University), Hiroto Ishiki, M.D. (Department of Palliative Medicine, National Cancer Center Hospital), Tetsuji Iriyama, M.D. (Specially Appointed Assistant Professor, Department of Internal Medicine, Palliative Care, Medical Hospital, Tokyo Medical and Dental University), Keisuke Kaneishi, M.D., Ph.D. (Department of Palliative Care Unit, JCHO Tokyo Shinjuku Medical Center), Tomofumi Miura, M.D., Ph.D. (Department of Palliative Medicine, National Cancer Center Hospital East), Yoshihisa Matsumoto, M.D., Ph.D. (Department of Palliative Medicine, National Cancer Center Hospital East), Ayumi Okizaki, Ph.D. (Department of Palliative Medicine, National Cancer Center Hospital East), Yuki Sumazaki Watanabe, M.D. (Department of Palliative Medicine, National Cancer Center Hospital East), Yuko Uehara, M.D. (Department of Palliative Medicine, National Cancer Center Hospital East), Eriko Satomi, M.D. (Department of Palliative Medicine, National Cancer Center Hospital), Kaoru Nishijima, M.D. (Department of Palliative Medicine, Kobe University Graduate School of Medicine), Junichi Shimoinaba, M.D. (Department of Hospice Palliative Care, Eikoh Hospital), Ryoichi Nakahori, M.D. (Department of Palliative Care, Fukuoka Minato Home Medical Care Clinic), Takeshi Hirohashi, M.D. (Eiju General Hospital), Jun Hamano, M.D., Ph.D. (Assistant Professor, Faculty of Medicine, University of Tsukuba), Natsuki Kawashima, M.D. (Department of Palliative Medicine, Tsukuba Medical Center Hospital), Takashi Kawaguchi, Ph.D. (Tokyo University of Pharmacy and Life Sciences Department of Practical Pharmacy), Megumi Uchida, M.D., Ph.D. (Dept. of Psychiatry and Cognitive-Behavioral Medicine, Nagoya City University Graduate School of Medical Sciences), Ko Sato, M.D., Ph.D. (Hospice, Ise Municipal General Hospital), Yoichi Matsuda, M. D., Ph.D. (Department of Anesthesiology & Intensive Care Medicine / Osaka University Graduate School of Medicine), Yutaka Hatano, M.D., Ph.D. (Hospice, Gratia Hospital), Satoru Tsuneto, M.D., Ph.D. (Professor, Department of Human Health Sciences, Graduate School of Medicine, Kyoto University Department of Palliative Medicine, Kyoto University Hospital), Sayaka Maeda, M.D. (Department of Palliative Medicine, Kyoto University Hospital), Yoshiyuki Kizawa M.D., Ph.D., FJSIM, DSBPMJ. (Designated Professor and Chair, Department of Palliative Medicine, Kobe University School of Medicine), Hiroyuki Otani, M.D. (Palliative Care Team, and Palliative and Supportive Care, National Kyushu Cancer Center).

## Author Contributions

**Conceptualization:** Jun Hamano, Ayano Takeuchi, Masanori Mori, Isseki Maeda, Takashi Yamaguchi, Tatsuya Morita, Takuya Shinjo.

**Data curation:** Ayano Takeuchi, Masanori Mori, Yasuhiro Saitou, Takahide Yamaguchi, Nobuyuki Miyata, Masakatsu Shimizu, Ryo Yamamoto, Yousuke Kimura, Yoshiyuki Kamiyama, Yasuyuki Arai, Hiroshi Matsuo, Hideki Shishido, Kazushi Nakano, Tomohiro Nishi, Takuya Shinjo.

**Formal analysis:** Jun Hamano, Ayano Takeuchi, Naosuke Yokomichi.

**Funding acquisition:** Jun Hamano, Hiroka Nagaoka.

**Investigation:** Jun Hamano.

**Methodology:** Jun Hamano, Ayano Takeuchi, Masanori Mori.

**Project administration:** Jun Hamano, Masanori Mori, Tatsuya Morita, Takuya Shinjo.

**Resources:** Yasuhiro Saitou, Takahide Yamaguchi, Nobuyuki Miyata, Masakatsu Shimizu, Ryo Yamamoto, Yousuke Kimura, Yoshiyuki Kamiyama, Yasuyuki Arai, Hiroshi Matsuo, Hideki Shishido, Kazushi Nakano, Tomohiro Nishi, Takuya Shinjo.

**Supervision:** Jun Hamano, Hiroka Nagaoka, Naosuke Yokomichi, Isseki Maeda, Takashi Yamaguchi, Tatsuya Morita, Takuya Shinjo.

**Writing – original draft:** Jun Hamano.

**Writing – review & editing:** Jun Hamano, Masanori Mori, Naosuke Yokomichi.

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
