## [Decision Letter · Decision Letter 0]

15 Dec 2022

PONE-D-22-27287Comparison of survival times of advanced cancer patients with palliative care at home and in hospitalPLOS ONE

Dear Dr. Hamano,

Thank you for submitting your manuscript to PLOS ONE. After careful consideration, we feel that it has merit but does not fully meet PLOS ONE’s publication criteria as it currently stands. Therefore, we invite you to submit a revised version of the manuscript that addresses the points raised during the review process.

We look forward to receiving your revised manuscript.

Kind regards,

Kenji Fujiwara, Ph.D., M.D.

Academic Editor

PLOS ONE

Journal Requirements:

a) Did participants provide their written or verbal informed consent to participate in this study?

Additional Editor Comments:

Dear Dr. Jun Hamano.

The manuscript is a prospective cohort study about the comparison of palliative care in hospitals and homes. The article was reviewed by two reviewers one recommended major revision and the other recommend rejection. I think the manuscript is eligible to proceed to major revision and re-evaluation after the revision.

Best regards,

Kenji Fujiwara

Academic editor

Reviewers' comments:

Reviewer's Responses to Questions

**Comments to the Author**

1. Is the manuscript technically sound, and do the data support the conclusions?

Reviewer #1: Yes

Reviewer #2: No

2. Has the statistical analysis been performed appropriately and rigorously? 

Reviewer #1: Yes

Reviewer #2: No

3. Have the authors made all data underlying the findings in their manuscript fully available?

Reviewer #1: No

Reviewer #2: Yes

4. Is the manuscript presented in an intelligible fashion and written in standard English?

Reviewer #1: Yes

Reviewer #2: Yes

5. Review Comments to the Author

Reviewer #1: In this secondary data analysis, data from two multicenter, prospective

cohort studies of advanced cancer patients was compared regarding survival time. One cohort was home-based, the other hospital-based. A Cox regression analysis with adjusting for some of the potential confounders including symptoms and treatments during the stay was conducted.

Thank you for giving me the opportunity to review this interesting paper. I am reviewing this paper from the perspective of a clinician-scientist with clinical as well as scientific experience in home-based palliative care in Germany.

Overall, the paper is very interesting, well written and scientifically well conducted.

I have to general points of inquiry that need to be addressed in a revised version:

1. From my experience, PC patients with a burden of symptoms that requires visits at home, care more about symptom-control than they care about survival time. Still it is totally acceptable to center this paper on survival time, but it would be strengthened if more comments could be added on the quality of symptom control/burden of suffering (e.g. in the abstract). In my opinion, adequate symptom control and survival time should always be linked in PC and displayed in close connection to each other.

2. The baseline data shows that the timing of enrollment in the two different types of care may differ in a clinically significant way and may therefore have a big influence on “survival time”. Table 1 shows, that patients with a “markedly poor health” are much more frequent in hospital-based PC (13.4% vs. 7.3%) which is also the case for a bad “palliative care prognostic index” (39.0% vs. 19.9%). Most of these factors are controlled for in the cox-regression model, but some are necessarily missing. I was confused though, that the ECOG wasn’t included in the model. Papers that the authors have cited (e.g. Just et al 2021) have shown, that performance status is an important factor connected to survival time. Table 2 shows clinically relevant differences in the ECOG regarding the two cohorts with a much higher burden of disability in the hospital group. I would therefore like to ask the authors to recalculate their model including ECOG upon enrollment/first available time-point as an independent variable.

3. Reasons for a decreased survival time in hospitals over time may be due to “hospitalization hazards” such as hospital acquired infections (e. g. MSRA, ESBL, Noro-Virus, SARS-CoV-2, etc.). Does the data include information on complications during the hospital stay?

Reviewer #2: I have reviewed 2 previous versions of this manuscript for another journal.

My main concerns about the study remain the same:

The main rationale the authors have for the study is that in their previous study comparing home PC clients and hospital PC clients they did not adjust for symptom severity and medical treatments during care. In other words, they want to shift the interest from going from a description of clients of two services to exploring a causal question about whether one of the services (causally) lengthens or shortens survival. That would, after all, be the reason for controlling for potential confounders.

I think that, despite the efforts done by the authors in refining some of their analysis compared to an early version of the manuscript, it is impossible to answer this causal question with the data they have. There is so much suspicion of residual confounding and at the same time there is not really a qualitative criterion for causality (eg the Bradford-Hill criteria) that is addressed (plausibility would be a minimum). I just have too much doubt about the contribution this manuscript would make to the state of science (and would even present a misleading message).

The analysis perhaps most plausibly shows that those clients referred to home PC are different from those referred to hospital PC, not only in their symptom severity but also in several other characteristics that the study did not measure. Like in other places around the world, referrals to hospital PC most likely just happen later than referrals to home PC.

But this manuscript goes way beyond that modest claim and infers a causal effect of the type of service. The authors even claim that their study provides the best possible evidence (with an RCT being impossible). I disagree that the study does.

Another comment I made in the previous reviews (as an aside) also remains: while the authors claim that all the variables controlled for in their model are confounders, treatments received during care are a mediator as they would be the result of the care service they are referred to.

6. PLOS authors have the option to publish the peer review history of their article (what does this mean?). If published, this will include your full peer review and any attached files.

Reviewer #1: No

Reviewer #2: No

---

## [Author Response · Author response to Decision Letter 0]

3 Feb 2023

Response to Review Comments

Reviewer #1: 

1. From my experience, PC patients with a burden of symptoms that requires visits at home, care more about symptom-control than they care about survival time. Still it is totally acceptable to center this paper on survival time, but it would be strengthened if more comments could be added on the quality of symptom control/burden of suffering (e.g. in the abstract). In my opinion, adequate symptom control and survival time should always be linked in PC and displayed in close connection to each other.

Reply#1 We agree with this comment. Although unfortunately, our study could not indicate a clear association between the quality of symptom control or the burden of suffering and survival time. However, we added this fact and possible explanation in the Discussion section. (p.23, l.207 - 210)

2. The baseline data shows that the timing of enrollment in the two different types of care may differ in a clinically significant way and may therefore have a big influence on “survival time”. Table 1 shows, that patients with a “markedly poor health” are much more frequent in hospital-based PC (13.4% vs. 7.3%) which is also the case for a bad “palliative care prognostic index” (39.0% vs. 19.9%). Most of these factors are controlled for in the cox-regression model, but some are necessarily missing. I was confused though, that the ECOG wasn’t included in the model. Papers that the authors have cited (e.g. Just et al 2021) have shown, that performance status is an important factor connected to survival time. Table 2 shows clinically relevant differences in the ECOG regarding the two cohorts with a much higher burden of disability in the hospital group. I would therefore like to ask the authors to recalculate their model including ECOG upon enrollment/first available time-point as an independent variable.

Reply#2 As the reviewer pointed out, the timing of enrollment in the two different types of care may differ. Moreover, we agree with the reviewer's comment that the performance status as the independent variable is essential. Since the PiPS-A and Palliative prognostic index include the component of performance status, we exclude the ECOG as the independent variable in terms of multicollinearity.

3. Reasons for a decreased survival time in hospitals over time may be due to “hospitalization hazards” such as hospital acquired infections (e. g. MSRA, ESBL, Noro-Virus, SARS-CoV-2, etc.). Does the data include information on complications during the hospital stay?

Reply#3 As the reviewer pointed out, hospital acquired infections and other adverse event in hospital might affect the survival time in hospital. Since, we did not assess the reason of the fever, it is difficult to speculate the “hospitalization hazards”. Therefore, the detailed assessment of the “hospitalization hazards” is important to conduct high quality observational study mentioned in recent article (Hernán, M. A. (2021). Methods of Public Health Research - Strengthening Causal Inference from Observational Data. The New England Journal of Medicine, 385(15), 1345–1348. https://doi.org/10.1056/NEJMP2113319).

Reviewer #2

My main concerns about the study remain the same:

The main rationale the authors have for the study is that in their previous study comparing home PC clients and hospital PC clients they did not adjust for symptom severity and medical treatments during care. In other words, they want to shift the interest from going from a description of clients of two services to exploring a causal question about whether one of the services (causally) lengthens or shortens survival. That would, after all, be the reason for controlling for potential confounders.

I think that, despite the efforts done by the authors in refining some of their analysis compared to an early version of the manuscript, it is impossible to answer this causal question with the data they have. There is so much suspicion of residual confounding and at the same time there is not really a qualitative criterion for causality (eg the Bradford-Hill criteria) that is addressed (plausibility would be a minimum). I just have too much doubt about the contribution this manuscript would make to the state of science (and would even present a misleading message).

The analysis perhaps most plausibly shows that those clients referred to home PC are different from those referred to hospital PC, not only in their symptom severity but also in several other characteristics that the study did not measure. Like in other places around the world, referrals to hospital PC most likely just happen later than referrals to home PC.

But this manuscript goes way beyond that modest claim and infers a causal effect of the type of service. The authors even claim that their study provides the best possible evidence (with an RCT being impossible). I disagree that the study does.

Reply#1 We already mentioned what reviewer concerned in limitation section as “however, we could not conclude a causal relationship or clarify the scientific mechanisms between the type of palliative of care and the survival time of advanced cancer patients in this observational study.” (p.24, l.224 - l.226)

In addition, we changed the description of discussion from “our current study was novel in confirming that patients receiving home-based palliative care had longer survival times than those receiving hospital-based palliative care by adjusting for symptoms and treatment factors.” to “our current study was novel to imply that patients receiving home-based palliative care had longer survival times than those receiving hospital-based palliative care by adjusting for symptoms and treatment factors.” (p.22 l.199) 

Furthermore, we changed the description of discussion from “our large-scale prospective, multicenter study with multifactorial adjustments is considered to be the highest level of evidence available.” to “our large-scale prospective, multicenter study with multifactorial adjustments is considered to be the high level of evidence available. However, RCTs in this area will continue to be ethically challenging, so it is necessary to perform the high-quality observational studies proposed in a recent article38.” We cited new article for this sentence. (Hernán, M. A. (2021). Methods of Public Health Research - Strengthening Causal Inference from Observational Data. The New England Journal of Medicine, 385(15), 1345–1348. https://doi.org/10.1056/NEJMP2113319))

Another comment I made in the previous reviews (as an aside) also remains: while the authors claim that all the variables controlled for in their model are confounders, treatments received during care are a mediator as they would be the result of the care service they are referred to.

Reply#2 We already mentioned what reviewer concerned in limitation section as “First, we were unable to adjust for residual confounding factors affecting the choice of the type of palliative care and survival time such as the preferences of patients and their families, family support, the details of dose-response treatment, and spiritual well-being.” (p.23, l.221 – p.24, l.224), and “however, we could not conclude a causal relationship or clarify the scientific mechanisms between the type of palliative of care and the survival time of advanced cancer patients in this observational study.” (p.24, l.224 - l.226)

---

## [Decision Letter · Decision Letter 1]

7 Mar 2023

PONE-D-22-27287R1Comparison of survival times of advanced cancer patients with palliative care at home and in hospitalPLOS ONE

Dear Dr. Hamano,

Thank you for submitting your manuscript to PLOS ONE. After careful consideration, we feel that it has merit but does not fully meet PLOS ONE’s publication criteria as it currently stands. Therefore, we invite you to submit a revised version of the manuscript that addresses the points raised during the review process.

The article is about the comparison of palliative care between at-home and in-hospital. This multi-center prospective study tried to adjust the survival time between two groups with symptoms. The manuscript is controversial among reviewers because the comparison has a risk to neglect some confounders and the disease severity of the two groups does seem not the same. I partly agree with the opinion of previous Reviewer #2 who recommended the rejection, and I think that the conclusion is too strong and the conclusion should be more modest. Please reconsider the conclusion by checking the suggestion from reviewers.

We look forward to receiving your revised manuscript.

Kind regards,

Kenji Fujiwara, PhD, MD

Academic Editor

PLOS ONE

Journal Requirements:

Additional Editor Comments:

Dear Dr. Hamano.

The article is about the comparison of palliative care between at-home and in-hospital. This multi-center prospective study tried to adjust the survival time between two groups with symptoms. The manuscript is controversial among reviewers because the comparison has a risk to neglect some confounders and the disease severity of the two groups does seem not the same. I partly agree with the opinion of previous Reviewer #2 who recommended the rejection, and I think that the conclusion is too strong and the conclusion should be more modest. Please reconsider the conclusion by checking the suggestion from reviewers.

On the other hand, this manuscript is a challenging study to compare two treatment styles though not perfect in order to encourage the home-based palliative care that many patients want.

I added my minor concerns below.

Best regards,

Kenji Fujiwara

Minor concerns.

1. In the abstract, “at 45 home-based PC” sounds confusing. I recommend using a 45-home-based PC “services”, “centers”, or “units” in order to distinguish from patients' numbers.

2. In the abstract, I found meaningless indention. Please delete it.

A total of 2,998 patients were enrolled in both studies and 2,878 patients were analyzed; 988 patients receiving home-based PC and 1,890 receiving hospital-based PC.

The survival time of patients receiving home-based PC was significantly longer than that of patients receiving hospital-based PC for the Days Prognosis (estimated median survival time: 10 days [95% CI 8.1 – 11.8] vs. 9 days [95% CI 8.3-10.4], p=0.157), the Weeks prognosis (32 days [95% CI 28.9 -35.4] vs. 22 days [95% CI 20.3-22.9], p < 0.001), and the Months Prognosis, (65 days [95% CI 58.2 – 73.2] vs. 32 days [95% CI 28.9-35.4], p < 0.001).

3. In the Ethics statement in the submission system, please indicate the form of consent obtained (written/oral) or the reason that consent was not obtained (the data were analyzed anonymously)

4. In Figure 1, I could not read the words in the PDF file. Please check whether the high-resolution image was uploaded.

Reviewers' comments:

Reviewer's Responses to Questions

**Comments to the Author**

1. If the authors have adequately addressed your comments raised in a previous round of review and you feel that this manuscript is now acceptable for publication, you may indicate that here to bypass the “Comments to the Author” section, enter your conflict of interest statement in the “Confidential to Editor” section, and submit your "Accept" recommendation.

Reviewer #1: All comments have been addressed

Reviewer #3: (No Response)

2. Is the manuscript technically sound, and do the data support the conclusions?

Reviewer #1: Yes

Reviewer #3: Partly

3. Has the statistical analysis been performed appropriately and rigorously? 

Reviewer #1: Yes

Reviewer #3: I Don't Know

4. Have the authors made all data underlying the findings in their manuscript fully available?

Reviewer #1: No

Reviewer #3: Yes

5. Is the manuscript presented in an intelligible fashion and written in standard English?

Reviewer #1: Yes

Reviewer #3: Yes

6. Review Comments to the Author

Reviewer #1: Dear Authors,

thank you for your comments as well as the revised version. I still have one minor comment regarding the abstract (as this will be the most-read part of the paper). I would suggest to rephrase the comment section as in its current form it may be misunderstood as a generalization of the results that may not be supported by the evidence and its limitations:

current version:

Conclusion

Our study revealed that advanced cancer patients with a Weeks or Months prognosis

receiving home-based PC survived longer than those receiving hospital-based PC after

adjusting for symptoms and treatments.

Suggestion:

Conclusion

In this cohort of advanced cancer patients with a Weeks or Months prognosis, those

receiving home-based PC survived longer than those receiving hospital-based PC after

adjusting for symptoms and treatments.

Reviewer #3: I agree with the comments of reviewer 1 and 2.

Essentially, patients referred to home palliative care services and hospital palliative care services are inherently different. The differences cannot be fully accounted for by statistical adjustment, due to the presence of unmeasurable confounders. While many of the comments from reviewers 1 and 2 have been addressed by including statements in the limitation section, the main text, in my opinion, still overstates the conclusions that can be reasonably drawn from this study.

7. PLOS authors have the option to publish the peer review history of their article (what does this mean?). If published, this will include your full peer review and any attached files.

Reviewer #1: No

Reviewer #3: No

---

## [Author Response · Author response to Decision Letter 1]

17 Mar 2023

The institutional review boards of all participating services approved this study and main institutional review boards (Come Home study: University of Tsukuba, EASED study: Seirei Mikatahara General Hospital) approved the use of existing EASED data for secondary analysis, and to combine the data for analysis. Japanese law does not require individual informed consent from participants in a non-invasive observational trial such as the present study. Therefore, we used an opt-out method rather than acquiring written or oral informed consent; all patients could receive information on the study through the instructions posted on the ward or institutional website, and they had the opportunity to decline participation.

---

## [Editor Report · Decision Letter 2]

27 Mar 2023

Comparison of survival times of advanced cancer patients with palliative care at home and in hospital

PONE-D-22-27287R2

Dear Dr. Hamano,

We’re pleased to inform you that your manuscript has been judged scientifically suitable for publication and will be formally accepted for publication once it meets all outstanding technical requirements.

Kind regards,

Kenji Fujiwara, PhD, MD

Academic Editor

PLOS ONE

Additional Editor Comments (optional):

Dr. Hamano.

Thank you for re-submitting the manuscript and I appreciate the responses to our suggestions. I think the manuscript is eligible to be accepted.

Best regards,

Kenji Fujiwara

---

## [Editor Report · Acceptance letter]

4 Apr 2023

PONE-D-22-27287R2 

Comparison of survival times of advanced cancer patients with palliative care at home and in hospital 

Dear Dr. Hamano:

I'm pleased to inform you that your manuscript has been deemed suitable for publication in PLOS ONE. Congratulations! Your manuscript is now with our production department. 

Kind regards, 

on behalf of

Dr. Kenji Fujiwara 

Academic Editor

PLOS ONE